# Urban Air Quality in a Coastal City: Wollongong during the MUMBA Campaign

**Clare Paton-Walsh** [1,*], **Élise-Andrée Guérette** [1,2], **Kathryn Emmerson** [2], **Martin Cope** [2],
**Dagmar Kubistin** [1], **Ruhi Humphries** [1], **Stephen Wilson** [1], **Rebecca Buchholz** [1,3],
**Nicholas B. Jones** [1], **David W. T. Griffith** [1], **Doreena Dominick** [1], **Ian Galbally** [1,2],
**Melita Keywood** [2], **Sarah Lawson** [2], **James Harnwell** [2], **Jason Ward** [2], **Alan Griffiths** [1,4] and
**Scott Chambers** [1,4]

[1]  Centre for Atmospheric Chemistry, University of Wollongong, Northfields Avenue,
    Wollongong, NSW 2522, Australia; eag873@uowmail.edu.au (É.-A.G.); kubid@gmx.de (D.K.);
    Ruhi.Humphries@csiro.au (R.H.); swilson@uow.edu.au (S.W.); buchholz@ucar.edu (R.B.);
    njones@uow.edu.au (N.B.J.); griffith@uow.edu.au (D.W.T.G.); dd824@uowmail.edu.au (D.D.);
    Ian.Galbally@csiro.au (I.G.); agf@ansto.gov.au (A.G.); szc@ansto.gov.au (S.C.)
[2]  CSIRO Climate Science Centre, Oceans and Atmosphere, Aspendale, Victoria 3195, Australia;
    Kathryn.Emmerson@csiro.au (K.E.); Martin.Cope@csiro.au (M.C.); Melita.Keywood@csiro.au (M.K.);
    Sarah.Lawson@csiro.au (S.L.); James.Harnwell@csiro.au (J.H.); Jason.Ward@csiro.au (J.W.)
[3]  Atmospheric Chemistry Observations & Modeling Laboratory, National Center for Atmospheric Chemistry,
    Boulder, CO 80301, USA
[4]  ANSTO Institute for Environmental Research, New Illawarra Rd, Lucas Heights, NSW 2234, Australia
*   Correspondence: clarem@uow.edu.au; Tel.: +61-2-4221-5065

**Abstract:** We present findings from the Measurements of Urban, Marine and Biogenic Air (MUMBA) campaign, which took place in the coastal city of Wollongong in New South Wales, Australia. We focus on a few key air quality indicators, along with a comparison to regional scale chemical transport model predictions at a spatial resolution of 1 km by 1 km. We find that the CSIRO chemical transport model provides accurate simulations of ozone concentrations at most times, but underestimates the ozone enhancements that occur during extreme temperature events. The model also meets previously published performance standards for fine particulate matter less than 2.5 microns in diameter ($PM_{2.5}$), and the larger aerosol fraction ($PM_{10}$). We explore the observed composition of the atmosphere within this urban air-shed during the MUMBA campaign and discuss the different influences on air quality in the city. Our findings suggest that further improvements to our ability to simulate air quality in this coastal city can be made through more accurate anthropogenic and biogenic emissions inventories and better understanding of the impact of extreme temperatures on air quality. The challenges in modelling air quality within the urban air-shed of Wollongong, including difficulties in accurate simulation of the local meteorology, are likely to be replicated in many other coastal cities in the Southern Hemisphere.

**Keywords:** air quality; traffic pollution; industrial emissions; modelling

---

## 1. Introduction

There is a growing understanding of the impact of poor air quality on premature deaths worldwide [1], underlining the importance of accurate modelling of air quality in populated areas. Recent decades have seen great advances in our understanding of urban air pollution, through both widespread long-term air quality monitoring in populated regions and targeted intensive measurement campaigns [2]. Long term monitoring programmes, which involve a network of low-maintenance

routine sites sampling a few key pollutants, are required to build up an understanding of air quality trends and population scale exposure, which is a pre-requisite for inferring the health impacts of poor air quality [3]. Conversely, targeted intensive campaigns, measuring a broad range of gaseous species and aerosol properties, are required to investigate the complexity of secondary organic aerosol formation and its coupling with gas phase photochemistry. Such a comprehensive suite of measurements is generally beyond the scope of long term monitoring programmes.

In Australia, a number of air quality studies were conducted around the millennium with a focus on ozone formation [4–9], with later interest moving to aerosol formation and composition [10–13] and impacts of vegetation fires and smoke (e.g., [14–16]). Previous campaigns have often concentrated on characterising a distinct environment, such as the urban air-shed [10]; forested regions dominated by biogenic emissions [17,18] or remote coastal regions that are heavily influenced by marine aerosol [13,19,20]. However, most Australians live in coastal cities surrounded by regions of native vegetation, which are a complex combination of all of these environments. The air quality within these areas is generally better than in other cities of the world [21], but accurately predicting air quality is complicated by:

- low background concentrations and boundary condition problems,
- influences from sea-breezes and local topology (that need to be correctly modelled);
- the interaction of marine aerosols with urban primary and secondary pollutants; and
- the interaction of biogenic volatile organics with urban pollutants.

The Measurements of Urban, Marine and Biogenic Air (MUMBA) campaign was designed to provide a detailed characterisation of atmospheric composition within such a coastal city, that is impacted by a mixture of anthropogenic, biogenic and marine sources. Given the high prevalence of coastal cities globally, aspects of this study are likely to be relevant to understanding air quality in many other parts of the world [22–25].

In this paper, we present observations of key species associated with air quality measured during the MUMBA campaign: Carbon monoxide (CO), sulphur dioxide ($SO_2$), nitric oxide (NO); nitrogen dioxide ($NO_2$), ozone ($O_3$), and particulate matter less than 2.5 microns in diameter ($PM_{2.5}$) and less than 10 microns in diameter ($PM_{10}$). These species are all included in legislated standards for air quality in Australia and are continuously monitored in the area, but during MUMBA these measurements were supported by a range of additional observations. For instance, measurements of volatile organic compounds, such as benzene ($C_6H_6$) and toluene ($C_7H_8$), can provide more detailed information on source apportionment for the observed concentrations of legislated pollutants, helping distinguish local and remote traffic emissions from the industrial emissions. Measurements of biogenic volatile organic compounds, such as isoprene and monoterpenes, provide information on these important precursors for $O_3$ and secondary organic aerosol formation. The monitoring of atmospheric constituents at multiple nearby sites also provides some insights into the variability of atmospheric composition within the urban air-shed and can help identify sources that contribute to pollution events.

In addition, we evaluate the ability of the CSIRO Chemical Transport Model (C-CTM) [6] to accurately predict key atmospheric pollutants within a coastal environment typical of where the majority of Australians live.

## 2. Materials and Methods

### 2.1. The MUMBA Campaign

The Measurements of Urban, Marine and Biogenic Air (MUMBA) campaign was a collaboration between the University of Wollongong, the Commonwealth Scientific and Industrial Research Organisation (CSIRO), the Australian Nuclear Science and Technology Organisation (ANSTO) and GNS Science, New Zealand. MUMBA involved the deployment of more than twenty different instruments in a coastal site within the regional city of Wollongong in New South Wales, Australia.

The campaign took place during austral summer from 21 December 2012 to 15 February 2013 and included two days of extreme heat, with maximum temperatures of 40.4 °C on 8 January

and 42.4 °C on 18 January 2013. A detailed description of the MUMBA campaign has already been provided in Paton-Walsh et al., 2017 [26], and the dataset is available for public access at PANGAEA (http://www.pangaea.de/) [27]. Most instruments were deployed at the eastern campus of the University of Wollongong (34.397° S, 150.900° E) hitherto referred to as "the MUMBA site", where sampling was conducted at approximately 10 m above ground-level from a dedicated mast. Instrumentation relevant for this study that was located at the MUMBA site included a chemiluminescence analyser for NO and $NO_2$; an ultraviolet absorption analyser for $O_3$, a Fourier transform spectrometer for measurement of CO (as well as $CO_2$, $CH_4$ and $N_2O$), and a proton transfer reaction mass spectrometer for volatile organic compounds (VOCs), including toluene and benzene.

Trace gas amounts are reported as dry air mole fractions in $\mu mol\ mol^{-1}$ (abbreviated ppm) or $nmol\ mol^{-1}$ (ppb). Particulate matter pollutants ($PM_{2.5}$ and $PM_{10}$) are reported as mass concentrations ($\mu g \cdot m^{-3}$). Some of the measurements from the MUMBA site provide a greater level of precision and accuracy than those available from long term monitoring (e.g., CO is available at ppb precision rather than at 0.1 ppm).

We also present $SO_2$, $PM_{2.5}$ and $PM_{10}$ data from the Office of Environment and Heritage (OEH) air quality station at Wollongong (34.419° S, 150.886° E), ~2 km southwest of the main MUMBA site. $SO_2$ was measured using pulsed fluorescent spectrophotometry, $PM_{10}$ was measured with a tapered element oscillating microbalance and $PM_{2.5}$ was measured with a beta attenuation monitor [28]. $PM_{2.5}$ data from the OEH station is used in this study, since monitoring of particle concentrations at the main MUMBA site was only conducted for just over four weeks of the eight-week campaign. The OEH data span the whole campaign and provide a self-consistent dataset to explore air quality in Wollongong during the study period. Additional FTIR measurements of CO, $CO_2$, $CH_4$ and $N_2O$ were available from the University of Wollongong main campus (at 34.406° S, 150.897° E) [29].

Most of the major influences on atmospheric composition in different directions from the measurement sites are evident in the satellite image of the region (Figure 1):

- ocean to the east:
- forest to the west (including a steep escarpment and forested region beyond);
- industrial complex to the south (at Port Kembla).

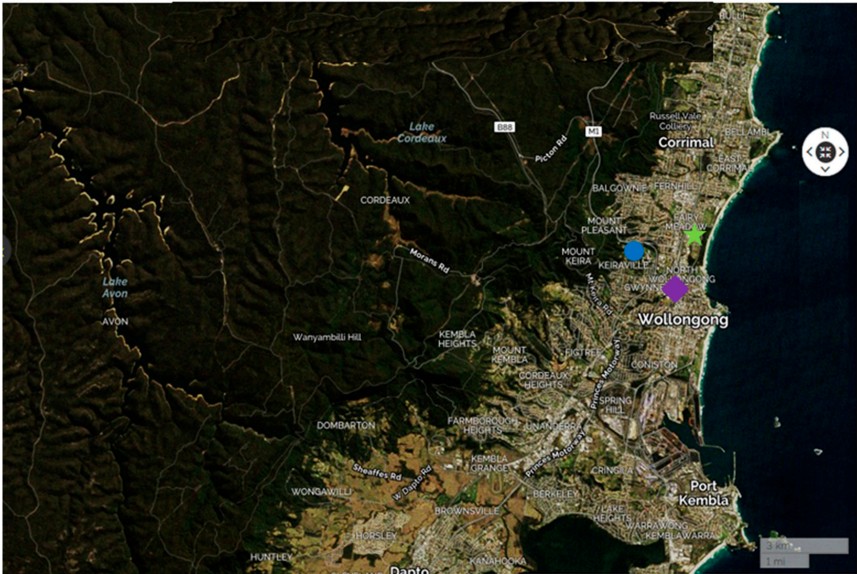

**Figure 1.** Satellite view of the region showing the main Measurements of Urban, Marine and Biogenic Air (MUMBA) site (green star), Wollongong Office of Environment and Heritage (OEH) Air Quality station (purple diamond) and the University of Wollongong (blue circle). Also visible is the large industrial area at Port Kembla and the extensive forested regions to the West. The image was created using the website: www.mapquest.com "© OpenStreetMap contributors".

Sydney is located 80 km to the north and has a population of approximately 5 million people. In the mornings, katabatic flow from the Blue Mountains region west of Sydney frequently pushes polluted air from the Sydney Basin east out over the ocean, such that the regular afternoon north easterly sea breezes may bring aged polluted air parcels from Sydney to the Wollongong area. During cold, still nights, katabatic drainage can also influence Wollongong directly, bringing air from the forested regions to the west into the urbanised coastal strip [30].

### 2.2. Emissions

Since air quality within a city is strongly influenced by local emissions, it is worthwhile considering the available data on these emissions. Major industrial sources of atmospheric pollutants are listed in Australia's National Pollution Inventory (NPI, see http://www.npi.gov.au/), along with an estimate of diffuse anthropogenic emissions (such as those from traffic). The New South Wales Environmental Protection Agency (NSW EPA) provides a more detailed inventory for the NSW Greater Metropolitan Region (GMR), however the most recent dataset that is publicly available is the '2008 Calendar Year Air Emissions Inventory for the Greater Metropolitan Region in NSW' [31]. Table 1 shows the equivalent mass of a number of key pollutants emitted to the atmosphere during the 57 days of the MUMBA campaign from the whole of New South Wales (NSW) and from the Wollongong region from these two inventories. The mass of emissions was calculated using the annual data for 2012/2013 from the NPI and the annual data for 2008 from the NSW EPA for these two regions and by scaling by the number of days (i.e., 57/365). The percentage of contributions to the total emissions of each pollutant from Wollongong in the NPI are given for the five most significant source categories. The availability of accurate emissions data is of paramount importance to successful air quality modelling and uncertainties in the emissions remain a significant contributor to the overall uncertainties in air quality modelling.

Despite its reputation as an industrialised city, Wollongong contributed 3.6% or less of the anthropogenic emissions of $PM_{2.5}$, $PM_{10}$, $NO_X$, $SO_2$ and VOCs listed in the NPI for New South Wales during 2012/2013 and 5% or less of anthropogenic emissions listed in the NSW EPA inventory for 2008. This is consistent with Wollongong's relative population, which is 3.9% of the total for New South Wales (292,000 residents of 7.52 million). The exception is CO: According to these inventories, Wollongong emits more CO per capita than the average for New South Wales. For CO, and all the other key pollutants listed in Table 1 except VOCs, "basic ferrous metal manufacturing" is the dominant category of emitters, which represents the steelworks and associated industries in the Port Kembla industrial area. The NSW EPA inventory for 2008 estimates four times as much CO is emitted within the Wollongong region than the NPI estimates. This is most likely the result of significantly reduced production at the steelworks and other industries around Port Kembla between 2008 and 2012/2013.

There can also be significant emissions of CO from forest fires in New South Wales [14], but during MUMBA, very few fires were close enough to potentially impact on the measurements. This is evidenced by low acetonitrile mole fractions throughout the campaign [32].

**Table 1.** Comparison of emissions within New South Wales (NSW) and Wollongong estimated in the National Pollution Inventory (NPI) for 2012/2013 and the NSW EPA inventory for 2008. Column 2 and 4 give the total NSW emissions in Giga-grams (Gg) for the 57 days of the MUMBA campaign from the 2008 NSW EPA inventory and the 2012/2013 NPI database respectively for each pollutant listed in Column 1. Column 3 and 5 show the emissions just from the Wollongong region in Giga-grams and as a percentage of the total NSW emissions from the 2008 NSW EPA inventory and the 2012/2013 NPI database respectively. Column 6 shows the five most significant anthropogenic source types within the Wollongong region in the NPI.

| Pollutant | NSW EPA Emissions (Gg) | Wollongong NSW EPA Emissions in Gg and (as % of NSW) | NPI NSW Emissions (Gg) | Wollongong NPI Emissions in Gg and (as % of NSW) | Breakdown of Anthropogenic Emission Sources from Wollongong | |
|---|---|---|---|---|---|---|
| $PM_{2.5}$ | 6.1 | 0.31 (5.0%) | 0.87 | 0.03 (3.6%) | Basic Ferrous Metal Manufacturing | 93.2% |
| | | | | | Coal Mining | 4.0% |
| | | | | | Electricity Generation | 1.7% |
| | | | | | Water Transport Support Services | 0.4% |
| | | | | | Water Supply, Sewerage and Drainage | 0.2% |
| $PM_{10}$ | 19 | 0.47 (2.4%) | 29 | 0.35 (1.2%) | Basic Ferrous Metal Manufacturing | 68.4% |
| | | | | | Solid fuel burning (domestic) | 11.3% |
| | | | | | Coal Mining | 9.1% |
| | | | | | Motor Vehicles | 6.0% |
| | | | | | Water Transport Support Services | 1.9% |
| CO | 151 | 84 (56%) | 171 | 21 (12%) | Basic Ferrous Metal Manufacturing | 80.4% |
| | | | | | Motor Vehicles | 16.1% |
| | | | | | Lawn Mowing | 1.3% |
| | | | | | Solid fuel burning (domestic) | 1.2% |
| | | | | | Lawn Mowing (public open spaces) | 0.3% |
| $NO_x$ | 50 | 1.8 (3.7%) | 50 | 1.7 (3.4%) | Basic Ferrous Metal Manufacturing | 56.4% |
| | | | | | Motor Vehicles | 34.2% |
| | | | | | Electricity Generation | 3.1% |
| | | | | | Railways | 2.0% |
| | | | | | Commercial Shipping/Boating | 1.9% |
| $SO_2$ | 45 | 1.4 (3.1%) | 35 | 0.79 (2.2%) | Basic Ferrous Metal Manufacturing | 94.8% |
| | | | | | Commercial Shipping/Boating | 1.5% |
| | | | | | Motor Vehicles | 1.3% |
| | | | | | Basic Chemical Manufacturing | 1.2% |
| | | | | | Railways | 0.6% |
| VOCs | 48 | 1.4 (2.8%) | 31 | 0.95 (3.1%) | Basic Ferrous Metal Manufacturing | 94.8% |
| | | | | | Motor Vehicles | 40.1% |
| | | | | | Domestic/Commercial solvents/aerosols | 16.2% |
| | | | | | Architectural Surface Coatings | 10.7% |
| | | | | | Solid fuel burning (domestic) | 8.6% |
| | | | | | Service stations | 4.9% |

*2.3. Regional Air Quality Modelling Using C-CTM*

We have used the CSIRO Chemical Transport Model (C-CTM) [6] to simulate atmospheric composition during the MUMBA campaign. The C-CTM is a modelling framework which simulates the emissions, transport and wet and dry deposition of chemical species in the atmosphere. Previously, the C-CTM has successfully been used in several air quality applications, including investigating smoke from biomass burning [33], shipping emissions [34] and biogenic emissions [35,36].

The complex topography surrounding the study region required that five successively higher resolution nested domains were used. The outer domain, which extends past the Australian borders to ensure that most of the significant remote sources of pollution are captured, has a spatial resolution of 80 km. Three smaller nested domains of resolutions 27 km, 9 km and 3 km encompass south eastern Australia, New South Wales and the Sydney Greater Metropolitan Region (GMR), respectively. The inner 1 km resolution domain extends 60 km × 60 km and is centred on the MUMBA campaign sites at Wollongong. The vertical resolution of all the layers in the model was 18 pressure contour levels from the surface to 200 hPa, and the model employed a 5 min chemical time step, outputting concentrations on an hourly basis.

Meteorology was provided to each model domain using the Conformal Cubic Atmospheric Model (CCAM, r2796, [37]), which uses initial conditions from the European Centre for Medium range Weather Forecasting Re-Analysis (ERA) Interim product to downscale the meteorological conditions to the required model resolution [38]. CCAM extends from the surface to 40 km in the vertical in 35 levels. Gas phase boundary conditions for ozone, methane, carbon monoxide, oxides of nitrogen, and seven VOC species, including formaldehyde and xylene, were taken from Cape Grim measurements [39], while those for the aerosol phase were taken from a global ACCESS-UKCA model run [40].

Anthropogenic emissions were taken from the '2008 Calendar Year Air Emissions Inventory for the Greater Metropolitan Region in NSW' [31], which includes 37 species. The inventory specifies on- and off-road mobile, aircraft, shipping, commercial, domestic and industrial point sources at a resolution of 1 km. Natural biogenic emissions of isoprene and monoterpenes come from the Australian Biogenic Canopy and Grass Emissions Model (ABCGEM, [35]). Monoterpenes are treated as a lumped species. Sea salt emissions are derived from a combination of a shore break mechanism [41], where the surf width is fixed at 20 m, and the wind driven parameterisation of Gong et al. [42]. Wind-blown dust emissions are derived using the algorithms of Lu and Shao [43]. The Global Fire Assimilation System (GFAS) at 10 km resolution is the source of wildfire emissions [44], which give the source location, as well as the plume height the smoke reached. The emission for particles and volatiles are speciated according to savannah vegetation conditions [32].

Organic species are lumped according to their carbon–carbon bonding type to suit the carbon bond 5 chemical mechanism (CB05) [45,46]. The mechanism has been extended to include updated toluene chemistry as per [45] and contains 65 gas phase species, 19 aerosol species and 172 reactions. The aerosol framework is via a two-bin sectional scheme, processing organic species by the volatility basis set [47] and processing inorganic species via ISORROPIA_II [48].

## 3. Results

*3.1. Criteria Pollutants during the MUMBA Campaign*

The CO time series measured at the main MUMBA site is shown in Figure 2 (each measurement takes three minutes). It is characterised by numerous enhancement events during which the CO far exceeds usual "background" values (seen by the thick undulating baseline of the plot). Enhancements above 500 ppb from clean Southern Hemisphere background levels around 50 ppb are not uncommon. However, a number of these events occur overnight and may therefore be the result of a build-up of local emissions within the nocturnal boundary layer. Note that the peak three minute values of approximately 2400 ppb (and peak hourly average values of 860 ppb) are well below the Australian National Environment Protection Measures (NEPM) ambient air quality standard for CO of

9000 ppb for 8 h [49]. Another feature is the significant variability of the underlying background values, ranging from ~50 ppb to ~100 ppb, suggesting large differences between regional-scale air-masses sampled during the campaign.

The potential impact of different air-masses sampled on Wollongong inhabitants can be better illustrated using a bivariate polar plot [50], which shows how a pollutant varies by wind speed and wind direction. Figure 3 is a bivariate polar plot for CO measured at the main MUMBA site throughout the campaign. Several distinct regions are evident, the most obvious being very high CO measured during southerly winds between 2 and 6 m·s$^{-1}$. Winds from this direction bring to the site air-masses that have passed over central Wollongong and the industrial area encompassing the Port Kembla steelworks south of the city centre. In contrast, the figure shows that easterly to south-south-easterly winds typically bring very low amounts of CO to the MUMBA site as the air-masses come from the Pacific Ocean.

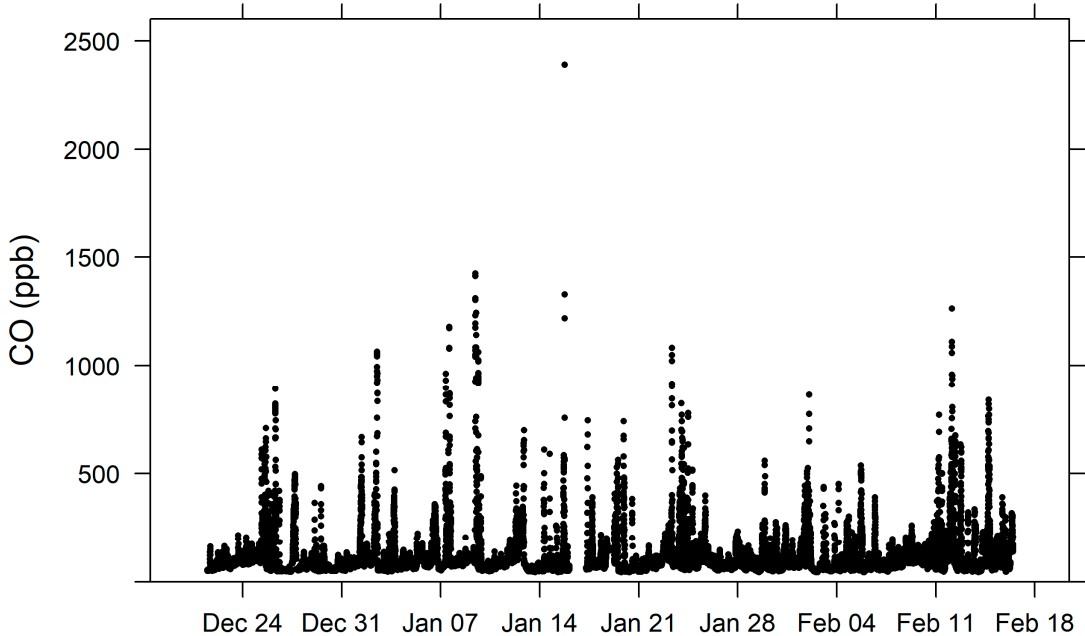

**Figure 2.** Time-series of three-minute average CO mole fractions (ppb) throughout the MUMBA campaign.

Carbon monoxide amounts in air from the north-east (that also comes off the ocean) are nearly double those from the south-south-east, indicating that the MUMBA site may be influenced by outflow from the Sydney basin, 80 km to the north. Elevated CO is also measured from the north-west in the direction of the nearest suburban shopping centre, a multilane road and local industrial sites (including a coke works and mining operations). In contrast, relatively low concentrations are seen from the south-west where there is a steep escarpment and eucalypt forests beyond.

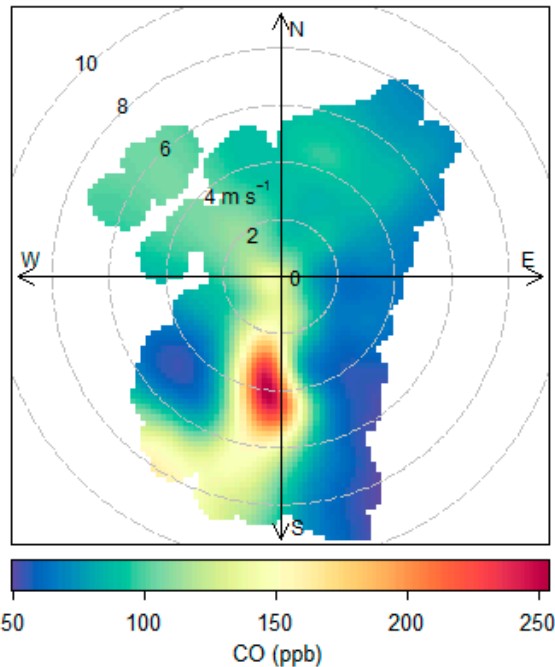

**Figure 3.** Bivariate polar plot showing CO mole fraction (ppb) variability as a function of wind speed (m·s$^{-1}$) and wind direction during the MUMBA campaign. Wind speed is represented by the concentric circles and wind direction is shown as compass directions, such that the shape of the coloured area illustrates the wind speeds and directions experienced during the campaign. The colour indicates the mean CO mole fraction measured under the corresponding wind conditions. All such figures in this study were created using the "openair" statistical tools for analysing air quality data using the software package R [51].

Table 2 shows the maximum and mean hourly concentrations of six key pollutants measured throughout the eight-week campaign, along with the relevant NEPM ambient air quality standard. Data are shown from the Wollongong OEH site and from the main MUMBA site (where available). Comparison of the average concentrations of CO, $NO_2$ and $O_3$ implies that the OEH site experienced somewhat more polluted air, with approximately 60% higher CO values, slightly higher $NO_X$ and lower $O_3$ on average than the main MUMBA site. The closer proximity of the OEH site to the Port Kembla industrial area and central urbanised area is implicated with average CO measurements (of 110 ppb) from the University of Wollongong in agreement with those at the main MUMBA site [52].

**Table 2.** Maximum and mean hourly average concentrations of key pollutants measured at the Wollongong OEH station throughout the MUMBA campaign and the relevant Australian Air Quality Standards. * Where standards apply over an averaging period of one day, values for daily averages are also given in parentheses.

| Pollutant | Australian Air Quality Standard NEPM | Maximum Hourly * (and Daily) Averages at Wollongong OEH Site during MUMBA | Mean Hourly Average at Wollongong OEH Site during MUMBA | Maximum Hourly Averages at Main MUMBA Site | Mean Hourly Averages at Main MUMBA Site |
|---|---|---|---|---|---|
| CO | 9000 ppb over 8 h | 1600 ppb | 180 ppb | 860 ppb | 110 ppb |
| $NO_2$ | 120 ppb over 1 h | 29 ppb | 5.7 ppb | 23 ppb | 5.2 ppb |
| $O_3$ | 100 ppb over 1 h | 66 ppb | 15 ppb | 54 ppb | 18 ppb |
| $SO_2$ | 200 ppb over 1 h | 18 ppb | 0.9 ppb | n/a | n/a |
| $PM_{10}$ * | 50 µg·m$^{-3}$ over 1 day | 185 * (47) µg·m$^{-3}$ | 23 µg·m$^{-3}$ | n/a | n/a |
| $PM_{2.5}$ * | 25 µg·m$^{-3}$ over 1 day | 48 * (16) µg·m$^{-3}$ | 7.3 µg·m$^{-3}$ | n/a | n/a |

* Target is no more than one day exceedance per year.

During the campaign there were no NEPM exceedances at Wollongong, with maximum values for most pollutants comfortably below recommended levels. In particular both CO and $SO_2$ mole fractions were typically one or two orders of magnitude lower than recommended limits, despite the presence of nearby industrial sources. $PM_{10}$ is the only pollutant that neared the recommended

maximum values, with a 24-h average concentration of 47 μg·m$^{-3}$ on the 18 January 2013, just below the NEPM ambient air quality standard of 50 μg·m$^{-3}$, averaged over one day.

The 18 January 2013 was the hottest day of the campaign and also the day with the highest PM$_{2.5}$ 24-h average concentration. The highest hourly averaged concentrations of both PM$_{10}$ and PM$_{2.5}$ were also observed on this day, although these peak values occurred at different times of day and under very different meteorological conditions. PM$_{2.5}$ peaked at 48 μg·m$^{-3}$ between 12:00 noon and 14:00 when hot north-westerly winds brought air parcels over the nearby eucalypt forests. PM$_{10}$ levels were relatively low at this time (averaging 18.8 μg·m$^{-3}$ over the 2 h), suggesting that secondary organic aerosol might be a significant source of the high PM$_{2.5}$ (see [52] for further discussion of this event). In contrast PM$_{10}$ peaked between 18:00 and 20:00 with hourly values of 185 μg·m$^{-3}$ and 148 μg·m$^{-3}$ from 18:00–19:00 and 19:00–20:00 respectively. PM$_{2.5}$ values were not available during the first hour, due to instrumental issues, but were relatively high (38 μg·m$^{-3}$) between 19:00 and 20:00. During these two hours a southerly change moved through, bringing strong gusty winds (averaging 2.4 m·s$^{-1}$ and 5.0 m·s$^{-1}$ respectively), suggesting a combination of raised dust and windblown sea-salt as the likely dominant sources of peak PM$_{10}$ levels.

Figure 4 shows PM$_{10}$, PM$_{2.5}$ and SO$_2$ polar bivariate plots of hourly concentrations measured from the OEH air quality station at Wollongong. The peak values measured on the 18 January 2013, associated with strong south-easterly winds, show up clearly in red in the PM$_{10}$ image, dominating the average values with these wind conditions during the campaign. The high PM$_{2.5}$ values on this day are less dominant in the polar bivariate plot of PM$_{2.5}$, with average values being highest during moderate to strong southerly winds. Nevertheless, enhancement in PM$_{2.5}$ accompanying stronger north-westerly winds that bring enhanced biogenic influences is also evident. The polar bivariate plot for SO$_2$ is dominated by a sole feature showing enhancement from southerly winds, which bring air parcels influenced by the Port Kembla industrial area. It is noticeable that the SO$_2$ enhancements are limited to a narrower range of wind directions than the PM$_{2.5}$, demonstrating that the source of fine particulate matter originates from a larger area than the industrial sources emitting SO$_2$ to the atmosphere.

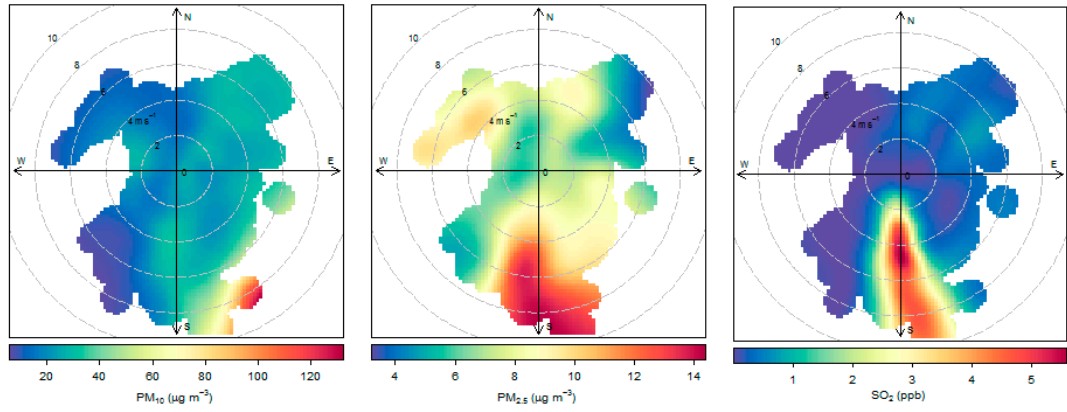

**Figure 4.** PM$_{10}$, PM$_{2.5}$ and SO$_2$ polar bivariate plots of hourly average measurements from the OEH air quality station at Wollongong during the MUMBA campaign.

*3.2. Traffic and Other Urban Influences*

The MUMBA site is significantly influenced by nearby traffic sources, which are expected to peak during the morning and evening rush hour times. High concentrations of NO, toluene and benzene are typically associated with traffic pollution although benzene is also emitted from the Port Kembla steel works. The short lifetime makes NO a clear marker for fresh traffic pollution, with toluene and benzene also good indicators of traffic pollution of different ages, because of their different atmospheric lifetimes (months for benzene and days for toluene) [53].

Figure 5 (top left) shows the variation of NO with wind speed and direction and has many of the same features that were evident in the CO bivariate polar plot. The dominant feature of this figure is the higher

levels that occur under westerly wind conditions with low speeds (0–2 m s$^{-1}$). This is largely attributable to the formation of stable nocturnal boundary layers (that can persist several hours after sunrise), which reduce vertical mixing and trap local emissions close to the source. The clear-sky and low gradient wind conditions that typically give rise to stable nocturnal conditions can also result in katabatic drainage from the Wollongong escarpment [30], which could explain why the high nocturnal NO concentrations appear to originate from the west. The strong NO signature with winds from the south again implicates the Port Kembla industrial area, whereas the lowest NO concentrations were observed with winds coming from the south-south-east. The suspected Sydney pollution coming from the north-east is very clear in the NO record. There is a road junction ~200 m to the north-east of the site, however the influence is less pronounced for low wind speeds (0–2 m·s$^{-1}$) than for higher wind speeds (2–7 m·s$^{-1}$). This is indicative of a more distant source and outflow from Sydney is the suspected cause.

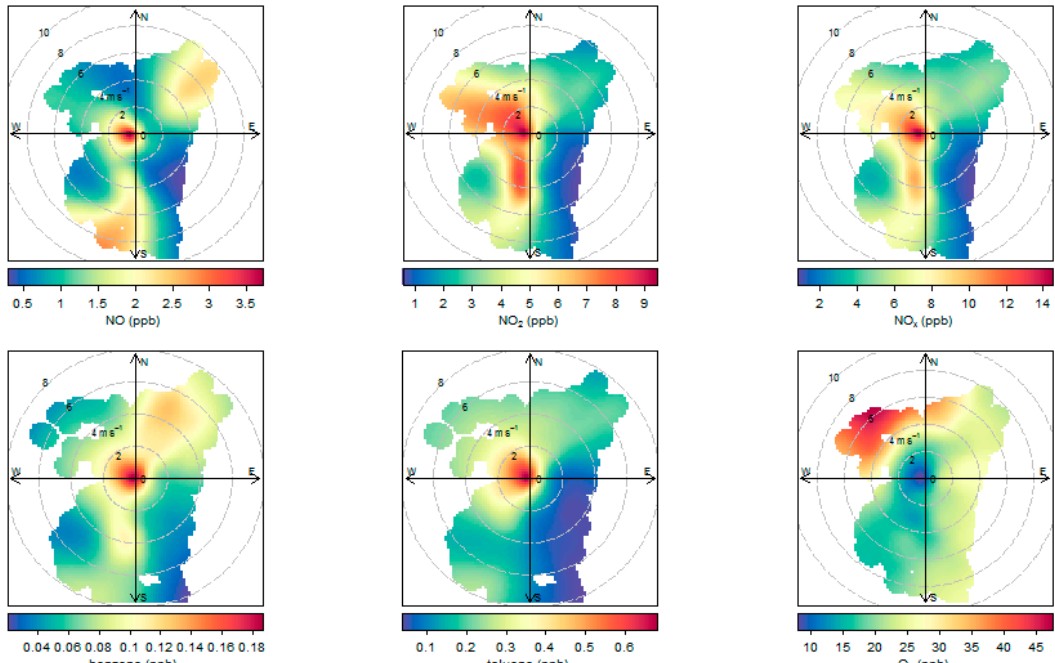

**Figure 5.** Bivariate polar plots showing how mole fractions (ppb) of different trace gases vary as a function of wind speed (m·s$^{-1}$) and wind direction at the main MUMBA site during the campaign.

The benzene and toluene polar bivariate plots (shown in the bottom left and centre of Figure 5 respectively), both show strong enhancements at low wind speeds, indicating local sources causing build-up within the nocturnal boundary layer. In contrast, low concentrations are associated with marine air from the south-east, and toluene concentrations decrease as the wind speed increases in all other directions, again indicating local traffic as the dominant source. Benzene shows stronger enhancements than toluene from the south (as it is emitted from the Port Kembla steel works) and also from the north-east, in the direction of outflow from the Sydney basin, consistent with aged pollution and the longer atmospheric lifetimes of benzene compared to toluene [53].

The O$_3$ polar bivariate plot (Figure 5 bottom right) shows the lowest values with low wind conditions, due to O$_3$ being titrated out by NO within the stable nocturnal boundary layer. Ozone is also low from the south (probably as a result of titration by NO$_x$ emissions from Port Kembla), and usually displays average concentrations from all easterly directions. Exceptions occurred for a few hours on 1 January and 22 January 2013 when winds were from the north-north-east and O$_3$ exceeded 40 ppb. These events are likely attributable to the outflow of polluted air from the Sydney basin returning on the sea-breeze. While there were no O$_3$ exceedances recorded during the MUMBA campaign, the highest values occurred during the two extreme hot days (8 and 18 January 2013), with

$O_3$ peaking at 48 ppb and 54 ppb respectively, when there were relatively strong winds from the north-west, where there are the largest biogenic influences [54].

## 4. Discussion

### 4.1. Comparison of C-CTM Modelled Air Quality Indicators with Measurements

Here we compare measurements of $PM_{10}$ and $PM_{2.5}$ from the Wollongong OEH site and $O_3$, $NO_X$, CO, toluene, isoprene and monoterpenes from the main MUMBA site to the predictions from the chemical transport model C-CTM. The comparison is of average hourly values from the measurement sites with average hourly C-CTM values within a 1 km by 1 km grid box that encompasses each measurement location. The model is not expected to be capable of simulating all the short term variability in the measurement data resulting from small-scale, localised events, but should have better skill at reproducing fluctuations in the background data that represent changes in air-mass at the scale of the model. Basic skill statistics for the comparison of C-CTM 1 km by 1 km grid output to point measurements of $PM_{10}$ and $PM_{2.5}$ at the NSW OEH station at Wollongong, and measurements of CO, $O_3$, $NO_X$ and toluene, isoprene and monoterpenes from the main MUMBA site are given in Table 3. Time-series showing model predictions and observations for all these species are shown in Figure 6.

Overall the model appears to capture the majority of fluctuations in $PM_{2.5}$ quite well (see top panel of Figure 6). In particular, C-CTM captures the $PM_{2.5}$ peak on 8 January 2013, which was the first of the two extreme temperature days of the MUMBA campaign (when temperatures reached over 40 °C). C-CTM also correctly predicts a peak in $PM_{2.5}$ on the other extreme temperature day on the 18 January 2013, although the timing and magnitude are not perfectly captured. Despite appearing to reproduce the observations reasonably well, the correlation coefficient is low at only 0.25. This poor correlation is likely to be driven by the inability of C-CTM to correctly predict the timing and magnitude of short time-scale peak $PM_{2.5}$ concentrations, since the longer time-scale fluctuations are reproduced quite well and the correlation coefficient is much higher (0.54) for daily averages. The model performs a little better on the other three basic skills tests that we applied, with 62% of model hourly $PM_{2.5}$ simulations within a factor of two of the observations, a normalised mean gross error of 61% and a relatively small positive normalised mean bias of 12% (with model overestimating the measurements).

Emery et al. [55] suggest performance goals (the level of accuracy that is considered to be close to the best a model can be expected to achieve) and criteria (the level of accuracy that is considered to be acceptable for modelling applications) based upon analysis of the performance of air quality models in the USA. Modelling air quality in coastal cities in the Southern Hemisphere is likely to be more challenging, due to the increased impact of the chosen boundary conditions, on the low ambient concentrations and the difficulties in modelling sea-breezes correctly. Nevertheless, C-CTM's performance meets the goal defined by Emery et al. [55] for daily $PM_{2.5}$ that the normalised mean gross error should be less than 35% and that the normalised mean bias should be less than ±10%. The model also meets the criteria that the correlation coefficient should be greater than 0.40.

**Table 3.** Basic skill statistics for CSIRO Chemical Transport Model (C-CTM) 1 km x 1 km grid to point measurements of $PM_{10}$ and $PM_{2.5}$ at the NSW OEH Air Quality station at Wollongong, and measurements of CO, $O_3$, $NO_X$, toluene, isoprene and monoterpenes at the main MUMBA site. Data is hourly unless otherwise specified.

| Pollutant | Proportion of Data within a Factor of 2 of Observations | Normalised Mean Gross Error | Normalised Mean Bias | Correlation Coefficient (r) |
|---|---|---|---|---|
| $PM_{2.5}$ | 0.62 | 0.61 | 0.12 | 0.25 |
| $PM_{2.5}$ daily | 0.96 | 0.34 | 0.10 | 0.54 |
| $PM_{10}$ | 0.58 | 0.50 | −0.43 | 0.37 |
| CO | 0.64 | 2.0 | 1.52 | 0.27 |
| $NO_x$ | 0.53 | 0.62 | −0.37 | 0.36 |
| toluene | 0.34 | 0.73 | −0.69 | 0.35 |
| ozone | 0.87 | 0.31 | 0.04 | 0.60 |
| isoprene | 0.29 | 1.33 | 0.62 | 0.63 |
| monoterpenes | 0.38 | 1.04 | 0.37 | 0.43 |

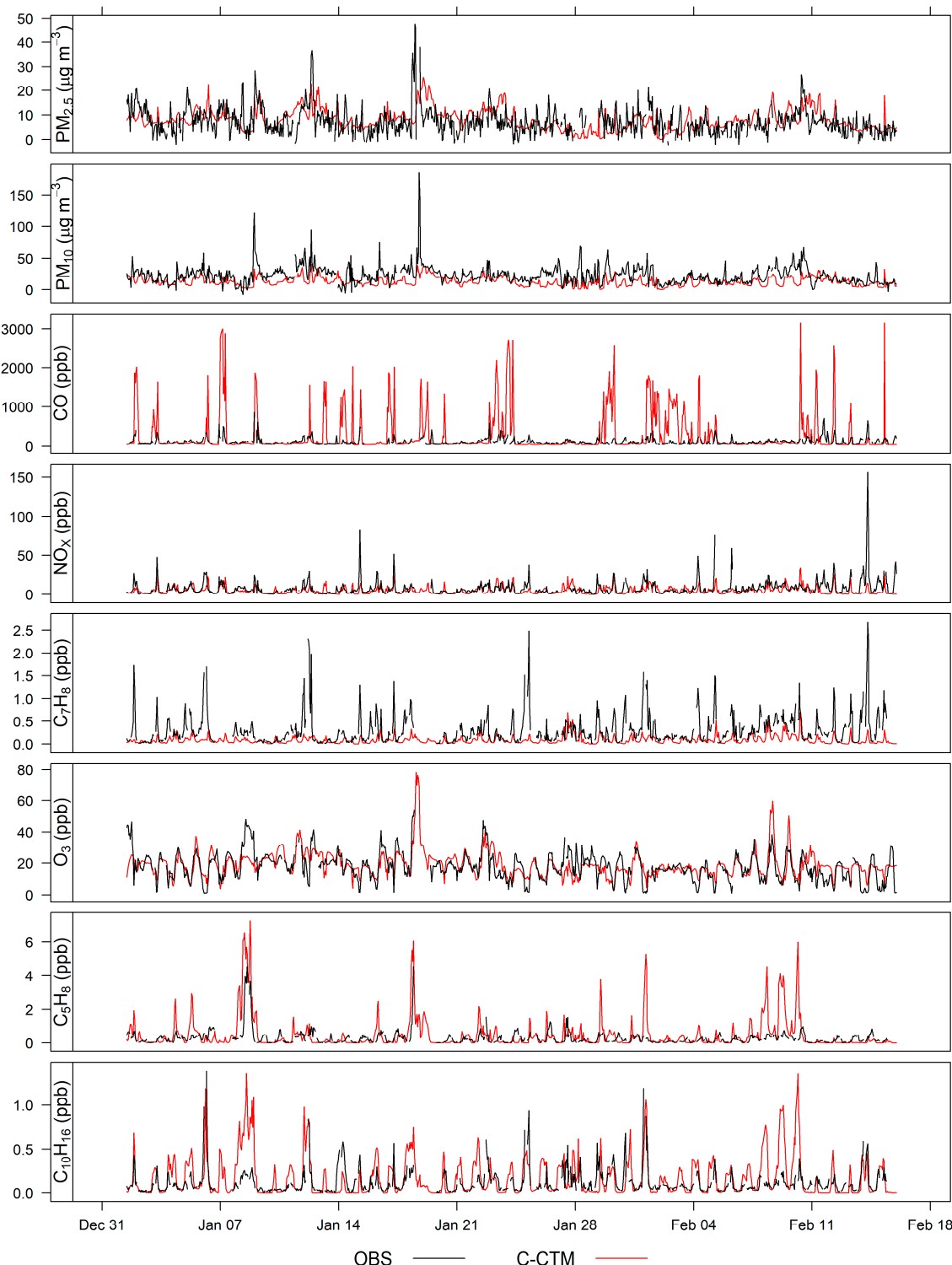

**Figure 6.** Time-series of C-CTM simulations at 1 km × 1 km resolution plotted with the corresponding point measurements of PM$_{2.5}$ and PM$_{10}$ at the NSW OEH Air Quality station at Wollongong, and measurements of CO, NO$_X$, toluene, O$_3$, isoprene and monoterpenes at the main MUMBA site. Note that only the overlap period is shown: The model simulations start on 1 January 2013 and the observations run until 15 February 2013.

Time-series of measured and modelled $PM_{10}$ are shown in the second panel of Figure 6, and the correlation coefficient is 0.37 with 58% of modelled data with a factor of two of the observations (Table 3). Despite C-CTM again capturing most of the fluctuations in baseline amounts of $PM_{10}$, there are several large peaks in the observations that are missed in the model, including peaks on the two extreme temperature days of the 8 and 18 January 2013. Statistically, C-CTM's performance shows a better correlation coefficient for $PM_{10}$. As Emery et al. [55] do not recommend updated benchmarks for $PM_{10}$, the model skill for $PM_{10}$ is evaluated using the benchmarks defined by Boylan and Russell, [49]. C-CTM meets both their model performance goals for particulate matter with a mean fractional bias smaller than $\pm$ 30% (−27.5%) and a mean fractional error smaller than 50% (37.8%).

From the third panel of Figure 7, the modelled CO appears to be vastly over-estimated, with a normalised mean bias of 1.52. However, the model captures the background fluctuations of CO well and the mismatch is dominated by times when there are peaks in CO concentration, predominantly as a result of winds from the south (in the direction of Port Kembla). C-CTM uses the NSW EPA emissions inventory for 2008 and (as discussed above), there was a significant reduction in industrial activity at Port Kembla between 2008 and 2012/2013, with the NPI emissions estimate for this latter time frame being only one quarter of the value assumed in C-CTM. Since C-CTM accurately simulates the baseline fluctuations in CO concentrations at the MUMBA site, the bias appears likely to be due to inaccurate estimates of the magnitude of local emissions. This result underlies the crucial importance of access to accurate emissions inventories for reliable air quality forecasting.

In contrast, $NO_X$ values display a significant negative bias, with the model underestimating mean $NO_x$ levels by 37% and, as in the case of $PM_{10}$, this appears to be driven by the model not capturing the magnitude of peak concentrations correctly. In fact, the model is missing the largest peak values in $NO_x$ and toluene. The correlation coefficient for $NO_x$ is very similar to that of toluene (0.36 and 0.35 respectively), suggesting that the traffic influence at the site is only partially replicated by the 1 km by 1 km C-CTM simulation. There is also a very significant underestimate of toluene in the model with a negative bias of 69%, with inaccuracy of the emission inventory again the most likely cause.

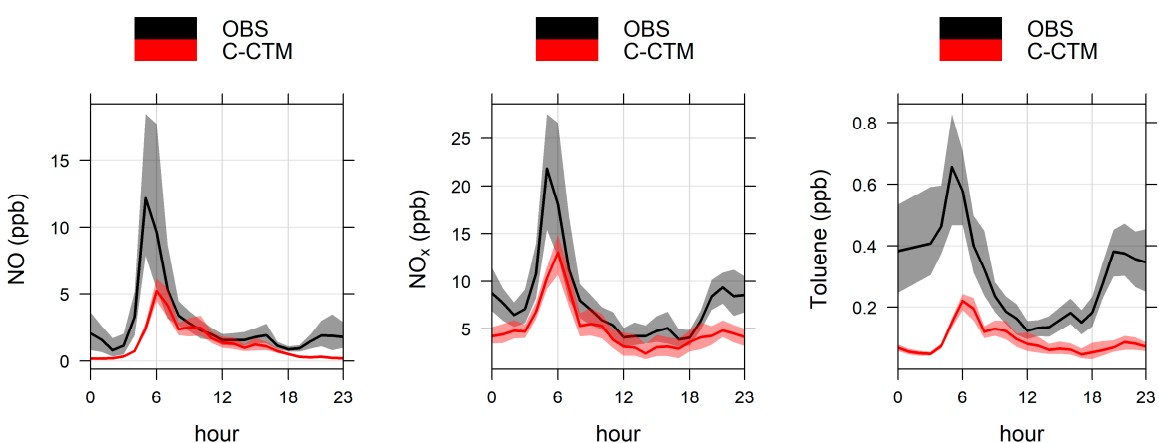

**Figure 7.** Average daily variations of mean mole fractions of NO, $NO_x$ and toluene during the campaign. Observations are shown in black and simulations using C-CTM are shown in red, with the dark central line representing the mean and the shaded area the 95% confidence interval.

Figure 7 (left) compares mean measured and modelled NO (ppb) over a composite diurnal cycle for the entire campaign. Measured NO pollution was dominated by a morning peak centred around 05:00 Australian Eastern Standard Time (AEST), which is equivalent to 06:00 local time with C-CTM modelled values for the 1 km by 1 km area containing the MUMBA site peaking at about half the measured values and one hour later. The observations are also significantly more variable (see shading) than the modelled values. The approximate timing would be expected for peaks caused by morning traffic pollution combined with a persistent stable nocturnal boundary layer, although the observed

peak is one hour earlier than the peak in morning traffic from local traffic counters [56]. There could also be a contribution from rapid photolysis of HONO that is present in a transported aged pollution plume [57], but given the relatively pristine background air, this contribution is likely to be small. Similar peaks with a 1-hour delay for the modelled values were seen in the $NO_X$ and toluene plots (see Figure 7 centre and right). The cause of the one hour offset between measured and modelled peaks is unclear when considering this data in isolation. This will be discussed in more detail in Section 4.2, which looks at data from multiple instruments at multiple sites.

The $NO_X$ concentrations are approximately double the NO concentrations during this morning peak, indicating roughly equal amounts of NO and $NO_2$. Freshly emitted $NO_X$ from traffic sources will be predominantly in the form of NO, but will be rapidly titrated by $O_3$ to form $NO_2$, such that equal amounts of NO and $NO_2$ is indicative of a relatively fresh pollution source [58]. The ratio of NO to $NO_2$ is captured well in the C-CTM data.

$NO_X$ emissions from the Port Kembla industrial area are also likely to impact the site, however the diurnal cycle of toluene (PTR-MS mass 93), which is predominantly from vehicular emissions, shows all the same features as the diurnal cycle of $NO_X$, confirming traffic as the most likely source of the morning NO peak.

A weaker signal in concentrations would be expected for the afternoon/evening rush-hour, since the traffic is both more dispersed in time and the air-shed is more vertically mixed (as the planetary boundary layer is normally deeper than in the morning). As expected, the afternoon rush-hour signals are very weak, with small peaks seen in NO, $NO_X$ and toluene at around 16:00 EST, which is equivalent to 17:00 local time. There are further night-time peaks in NO, $NO_X$ and toluene, most likely caused by local evening traffic emissions trapped in the re-established nocturnal boundary layer, with NO accounting for only approximately one quarter of the $NO_X$ peak, since the production of NO from the photolysis of $NO_2$ is shut off at sunset. Night time $O_3$ values are typically 10–15 ppb, so there is sufficient $O_3$ to convert NO to $NO_2$. The presence of a peak in NO is thus further evidence of a fresh source of $NO_X$ and anti-correlation of $NO_X$ and $O_3$ throughout the campaign is expected due to the titration of $O_3$ by NO.

C-CTM simulates $O_3$ at the site quite well (see the sixth panel of Figure 6), with 87% of values within a factor of 2 of the observations. The model also meets the criteria recommended by Emery et al. [55] that the correlation coefficient should be greater than 0.5 (0.6) and the recommended goal that the normalised mean bias should be less than 5% (4%). This may seem surprising given the large underestimation of toluene in a VOC-limited environment (the formaldehyde to $NO_X$ ratio averaged 0.3 over the campaign [26]), however, anthropogenic emissions of VOCs are relatively low in this region, making biogenic VOCs important to the overall budget. This version of C-CTM uses the Australian Biogenic Canopy and Grass Emissions Model (ABCGEM, [59]) for biogenic emissions of isoprene and monoterpenes. The simulated isoprene and monoterpenes (last two panels of Figure 7) are overestimated by 62% and 37%, potentially offsetting the underestimate of toluene and allowing a good $O_3$ simulation. The biases for isoprene and monoterpenes are much smaller than those previously reported using MEGAN2.1 [36,60], illustrating improved ability to capture biogenic emissions in south-eastern Australia. C-CTM misses the increased $O_3$ on the 8 January 2013 (the first of the extreme temperature days) but appears to capture the event on the 18 January. Although only the start of the event was measured at the main MUMBA site (as all instruments were turned off on this day, due to the inability of the air-conditioning to cope in the extreme heat [26]), the observed $O_3$ at the OEH site showed peak values as predicted by C-CTM. It is important to understand the likely impacts on air quality of these extreme temperature events, as they are likely to occur more frequently in the future, due to changes in climate ([54,61]).

We conclude that C-CTM provides a good prediction of air quality indicators in the Wollongong region especially for $PM_{2.5}$ and $O_3$. These are the two most important pollutants for human health in the region, due to a combination of their toxicity and the frequency that their concentrations exceed the

regulatory standards [21]. Other pollutants are also simulated with reasonable skill by C-CTM, with improvements to the emissions inventory for CO and toluene likely to yield the greatest improvements.

*4.2. Insights from Observations Made at Nearby Sites and Using Multiple Pollutants*

One aspect of the drivers of air quality in Wollongong is the high concentration of polluting industrial activities that are located in a single area (Port Kembla). For example, 80% of CO emissions in the region come from basic ferrous metal manufacturing at Port Kembla (see Table 1). This strong point source will impact a given observation site (or not) depending on wind conditions. The impact of this can be illustrated by plotting the CO concentrations that are measured simultaneously at different sites within the region, as is done in Figure 8 for CO measured at the University of Wollongong and at the main MUMBA site.

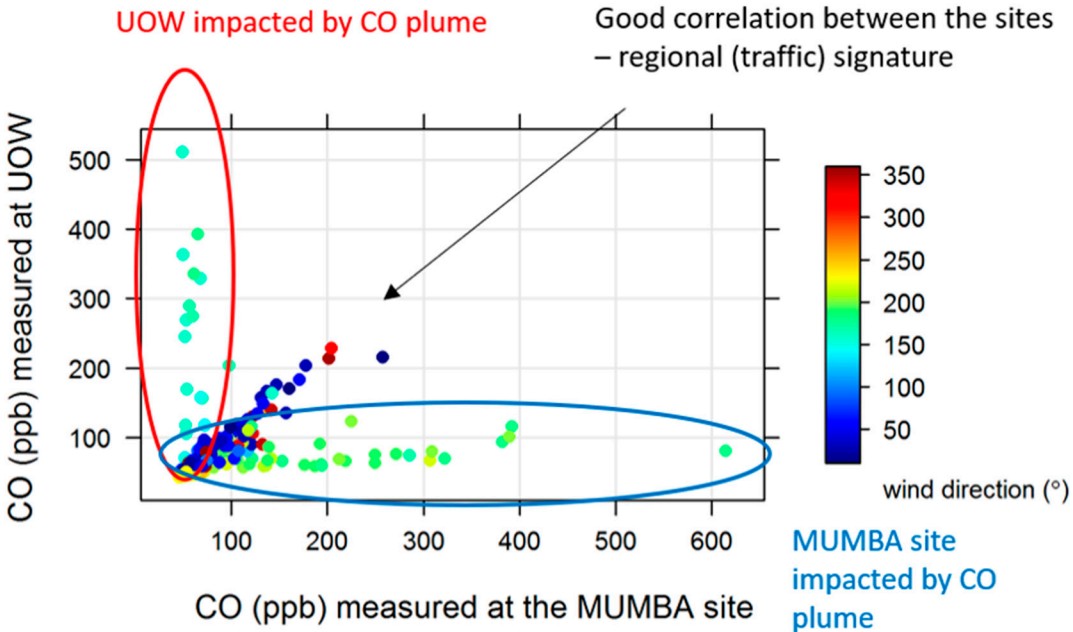

**Figure 8.** CO measured at the University of Wollongong (UOW) site plotted against coincident CO measured at the main MUMBA site. The plot is coloured by wind direction measured at the MUMBA site.

Figure 8 demonstrates that the CO plume from Port Kembla impacts one site or the other, depending on wind direction, with three distinct correlations patterns in the data:

- High CO at the University of Wollongong and low CO at the main MUMBA site with wind directions around 150°;
- High CO at the main MUMBA site and low CO at the University of Wollongong with wind directions around 200°;
- Strongly correlated CO data at both sites with all other wind directions.

This illustrates that the plume from Port Kembla impacts the MUMBA site only when the wind is from a narrow range of wind directions. If the modelled wind direction is slightly biased, there will be missed (or extraneous) CO events in the modelled time-series. This explains the poor correlation between the model and the observations (r = 0.27). Figure 8 also shows that there is another CO source that is measured at both sites regardless of wind direction. Examining the toluene concentrations during these events shows that the high CO plume from Port Kembla is toluene-poor, whereas the regional signature has toluene in it, pointing to traffic as the 'non-plume' CO source. (see Appendix A Figure A1).

If we consider $NO_x$ amounts, as well as CO and toluene, we get a more detailed idea of the local industrial and traffic signatures. Figure 9 shows $NO_x$ plotted against CO at the MUMBA site, coloured

by the amount of toluene present. From this, we can see that the MUMBA site is impacted by three types of event:

- The plume from Port Kembla with high CO, some $NO_x$ and low toluene;
- High toluene events with medium range enhancements in CO and $NO_x$;
- Events when CO, $NO_x$ and toluene are all high.

Considering other aspects of these events, we can see that the Port Kembla plume impacts the site during southerly winds and only at higher wind speeds, whereas the other two event types occur at low wind speeds. The high toluene events with medium range enhancements in CO and $NO_x$ occur at all hours of the night but not during the day, with winds anywhere from the south, west or north. These events are most likely due to traffic pollution trapped in the nocturnal boundary layer. The final type of events when toluene, CO and $NO_X$ are all high, occur at low wind speeds from the west to south-west in the early morning (around 05:00) and at colder overnight temperatures. These events are also characterised by high $NO_x$ amounts measured at the OEH station, (which indicates regional events rather than very local events) and when most of the $NO_x$ is in the form of NO, indicating fresh emissions. These events are most likely caused by katabatic drainage when cold air descends through the forested escarpment and across a number of major roads, picking up traffic emissions from the early morning commuters. Plots showing $NO_x$ plotted against CO coloured by wind speed, wind direction and hour of the day are shown in Appendix A Figure A2.

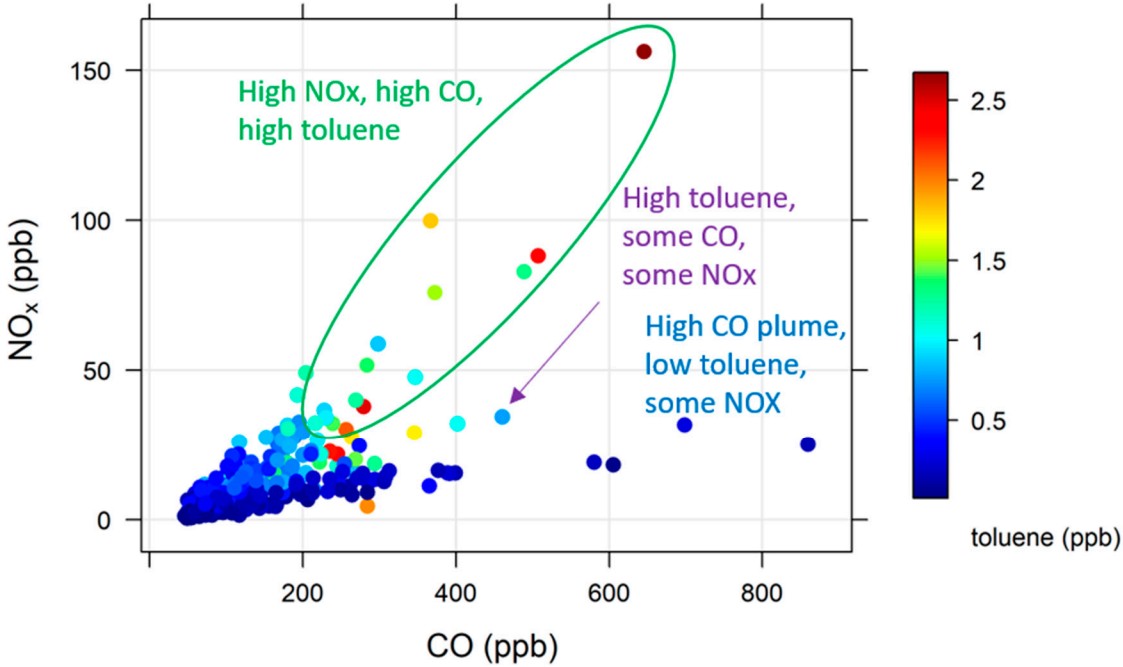

**Figure 9.** $NO_x$ plotted against CO measured at the main MUMBA site. Data points are coloured by toluene concentrations.

The high toluene, $NO_x$ and CO events were not captured by the C-CTM model as it does not reproduce the katabatic flow. Filtering out hours when the ratio of NO to CO is equal to or greater than 0.06 effectively removes these events from the dataset. Figure 10 shows the comparison of composite diurnal cycles of observed and modelled NO, $NO_X$ and toluene, with the katabatic flow events removed. Without these high concentration events in the observations, the agreement is much improved (compared to the whole dataset, shown in Figure 7), both in terms of timing and normalised mean bias. Indeed, there is no longer an hour offset in the timing of the morning peak values and the normalised mean bias improves to −0.25 (from −0.37). This demonstrates that the apparent different

timing was not due to any error in the timing of emissions, but due to the absence of the early morning katabatic flow events in the C-CTM data. The modelled toluene is still too low without the katabatic flow events, pointing to an underestimate in the emissions in the inventory.

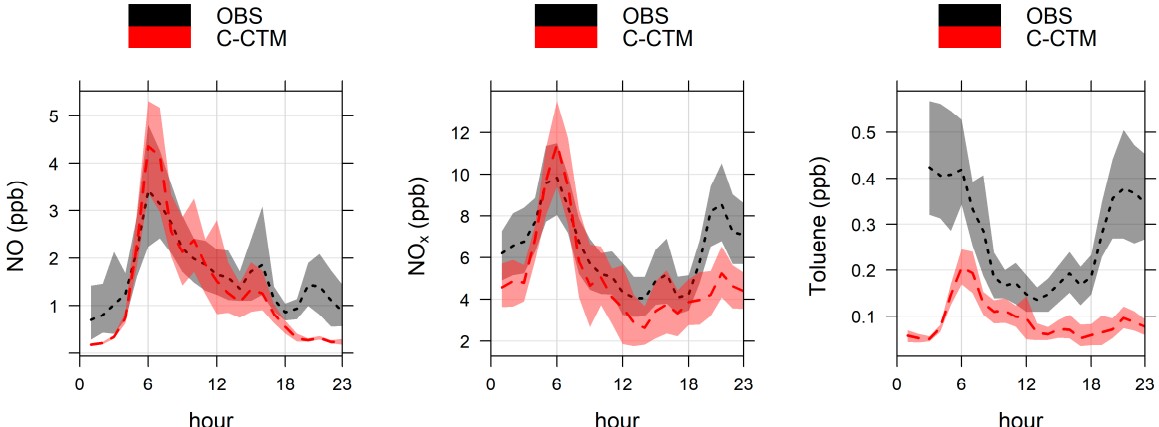

**Figure 10.** Average daily variations of mean mole fractions of NO, $NO_x$ and toluene during the campaign with all data with a ratio of NO to CO $\geq 0.06$ excluded (to remove the katabatic flow events). Observations are shown in black and simulations using C-CTM are shown in red, with the dotted central lines representing the mean and the shaded area the 95% confidence interval.

## 5. Summary and Conclusions

Urban air quality in the coastal city of Wollongong, New South Wales was shown to be highly variable during the MUMBA campaign. Key air quality indicators showed short-lived peak values typical of a polluted urban environment, as well as more sustained periods of Southern Hemisphere clean air background concentrations. This variability results from a complex mixture of different influences, including the local industrial and traffic emissions, marine air and biogenic influences from the surrounding vegetation. Despite significant industrial sources, including local steelworks, $NO_X$ concentrations were shown to be most influenced by traffic emissions and the air-shed is VOC-limited. Natural biogenic emissions play an important role in urban air quality in Wollongong, due to low anthropogenic emissions and the remoteness from other polluting regions. This is likely typical of coastal cities throughout Australia and in many other parts of the Southern Hemisphere (e.g., South America), where the largest population density occurs in urban areas bordered by forest and sea.

Air quality modelling in Southern hemispheric coastal cities is particularly challenging. This is because of the sensitivity of low ambient concentrations to the boundary conditions used in the model and because of the difficulties in correctly capturing the sea-breezes and local effects like the katabatic flow in the modelled meteorology. Despite these issues, the 1 km by 1 km C-CTM simulation generally agreed well with the observations of ambient air quality in Wollongong, especially for the criteria pollutants of greatest concern ($PM_{2.5}$ and $O_3$), with further improvements likely to be possible via access to an updated emissions inventory. Finally, improvements in our understanding of the impact of extreme temperatures on $O_3$ and $PM_{2.5}$ are needed, since these events are likely to be more frequent in future.

**Author Contributions:** Conceptualisation, C.P.-W., M.K., R.H.; methodology, C.P.-W., E.-A.G., D.K.; validation, E.-A.G., D.K., D.W.T.G.; formal analysis, E.-A.G., C.P.-W.; investigation, E.-A.G., D.G., C.P.-W., R.H., I.G., S.W., D.D., R.B., S.L., J.H., J.W., A.G., S.C., K.E., M.C.; data curation, E.-A.G., D.D.; writing—original draft preparation, C.P.-W., E.-A.G.; writing—review and editing, all authors.; visualisation, E.-A.G.; supervision, C.P.-W., S.R.W.; project administration, C.P.-W.; funding acquisition, C.P.-W.

**Funding:** This research was funded by Australia's National Environmental Science Program through the Clean Air and Urban Landscapes hub and from the Australian Research Council Discovery Project DP160101598. This research was also supported by the Australian Government Research Training Program (RTP) Scholarships.

**Acknowledgments:** The authors acknowledge the NSW Office of Environment and heritage for providing publicly available air quality data in Wollongong. The authors would like to thank all those from the University of Wollongong's Centre for Atmospheric Chemistry and CSIRO's Climate Science Centre group, who helped with the logistics of undertaking an extensive measurement campaign. Thanks are also due to Kids Uni and the Science Centre for their helpful support, and to David Carslaw (and all the statisticians who developed the relevant 'R' code) for public access to the excellent Openair package for analysis of air quality data.

**Conflicts of Interest:** The authors declare no conflict of interest.

## Appendix A

This appendix contains two additional figures that help illustrate the conclusions regarding the causes of different pollution events identified in the data.

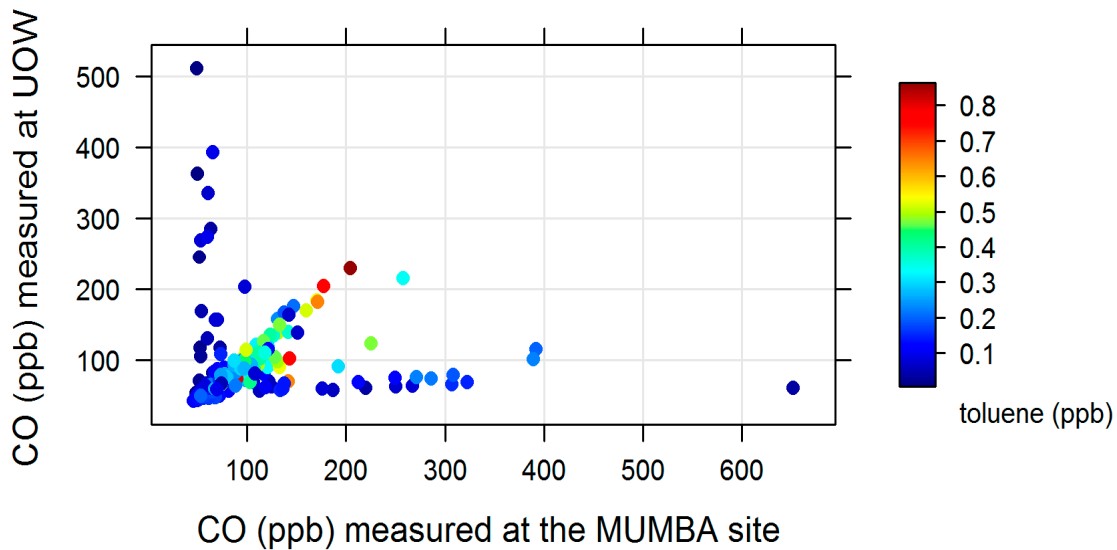

**Figure A1.** CO measured at the University of Wollongong (UOW) site plotted against coincident CO measured at the main MUMBA site. Data points are coloured by toluene concentration measured at the MUMBA site.

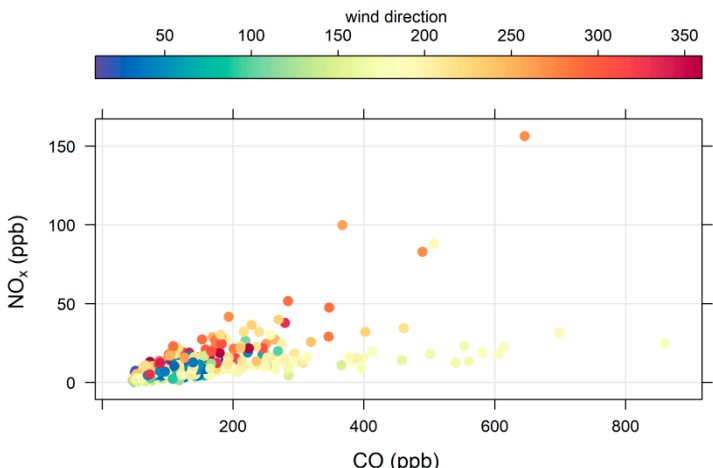

**Figure A2.** *Cont.*

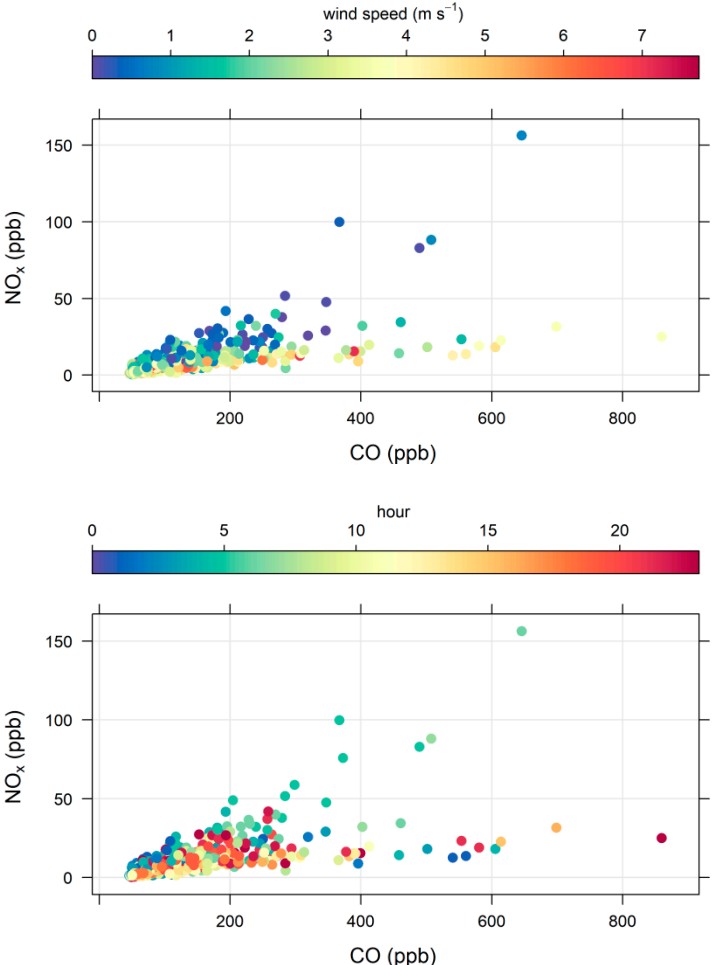

**Figure A2.** NO$_x$ plotted against CO measured at the main MUMBA site. The top plot is coloured by wind direction; the middle plot is coloured by wind speed and the bottom plot is coloured by the hour of the day.

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
