# Peer review of "Urban Air Quality in a Coastal City: Wollongong during the MUMBA Campaign"

_atmosphere, doi:10.3390/atmos9120500_

Round 1

Reviewer 1 Report

In the manuscript atmosphere-404851 the authors presented the results of an extensive study aimed at investigating of key species associated with air quality measured during the Measurements of Urban, Marine and Biogenic Air (MUMBA) campaign. A further aim was to evaluate the ability of the Australian air quality forecasting model C-CTM to accurately predict key atmospheric pollutants within a coastal environment. Selected key species (CO, SO2, NO, NO2, O3, PM2.5 and PM10) and a range of additional pollutants (to provide more detailed information on source apportionment for the observed concentrations the above-mentioned key-species) were measured during the MUMBA campaign. Key air quality indicators concentrations showed a certain variability (i.e., short-lived peak values - typical of a polluted urban environment - and more sustained spells of Southern Hemisphere clean air background), which was attributed to complex mixture of different influences, including the local industrial and traffic emissions, marine air and biogenic influences from the surrounding vegetation. Further, authors fund that the adopted chemical transport model provides accurate simulations of ozone concentrations at most times but underestimates the ozone enhancements that occur during extreme temperature events. The model also meets previously published performance standards for PM2.5, PM10. Authors also suggest that an enhancement of air quality simulations in this coastal city, can be made through more accurate anthropogenic and biogenic emissions inventories and better understanding of the impact of extreme temperatures on air quality.

Comments for the authors

This is a well-written, well-prepared, well-structured manuscript, based on an original and very interesting idea and on a well-designed and executed study. The introduction section provides sufficient elements to properly define the background and the problem statement. The adopted methodologies appear to be adequate for the stated purpose, and they are clearly presented in the manuscript. The manuscript doesn’t show relevant errors and omissions. In summary, this is a manuscript that deals with an interesting topic, addressed with a method that in some ways is innovative, that should be considered for publication on Atmosphere.

Author Response

We thank Reviewer 1 for their positive comments.

Since I cannot see that any changes were requested we have made no alterations to the manuscript in light of this first review

Reviewer 2 Report

1) In the paper, the authors present the interesting information on the following substances pollutants the air quality: 

   Monoxide (CO), sulfur dioxide (SO2), nitric oxide (NO); nitrogen dioxide (NO2), ozone (O3), and particulate matter

   less than 2.5 microns in diameter (PM2.5) and less than 10 microns in diameter (PM10);

2)The paper is not a very original topic.

   But it the paper with other published material, can add some important  informations on the subject area;

Author Response

We thank reviewer 2 for their comments and suggested edits to the paper.

This reviewer has suggested that we:

1). add 2 references to the introduction (which we have done along with 2 others) - thanks!

2). separate text from Table and Figure captions and place in the main text instead. (We have decided not to do this as we consider that the Table and Figures should be accompanied by all the information needed to fully understand them without needing to refer to the main text).

3). Fix typos where first letter is missing from captions (Done)

4.) Add a comment about the use of new technology (sensors) to help people avoid exposure to poor air quality and add a reference. (We have also decided not to do this as this paper does not mention or make use of new sensor technology and so this seems out of place as our final conclusion to this paper.)

Reviewer 3 Report

The manuscript “Urban Air Quality in a Coastal City: Wollongong during the MUMBA campaign” by Paton-Walsh et al. is an interesting study where the performance of C-CTM model was assess to model pollutants’ concentrations during MUMBA campaign. The manuscript is well written, and the results and procedures are well explained. Overall, I believe that this manuscript deserves to be considered for publication, after the authors review it regarding the few comments described below:  

The authors should use the template of the manuscript structure that is usually is used in Atmosphere Journal, namely: Introduction, Materials and Methods, Results, Discussion and Conclusions. Please follow the guidelines available at: https://www.mdpi.com/journal/atmosphere/instructions

Please, provide figures and tables with better quality (eg., Fig.1 )

Through all the manuscript, replace the use of “we” by some more formal expression, eg: instead of using (line 16) “We present findings (…)”, please use “This study aimed to present (…)” or similar.

Section 2 – How PM10 and PM2.5 levels were measured? With which instrumentation?

Line 158 – Supply a reference for low acetonitrile levels.

Table 1 – Use PM2.5 with 2.5 in subscript.

Sections 2 to 4 can be considered as “Methods” Section since section 5 starts to provide results of the MUMBA campaign.

Line 216 – reference for the guideline value of NEPM?

Figure 2 – yy axis should start at 0.

Table 2 – improve the units for PM10 and PM2.5: microg.m-3 (use the dot) and homogenize the units among the manuscript. The same for the velocity: use m.s-1 instead of ms-1

Line 272 – erase “.” after PM2.5

Line 277 – apply the same number of significant digits to average velocities of “strong gusty winds”.

Table 3 – line 371 – at its capton, it is missing a “B” in “basic”. Similar is happening to other captions (almost all Figures). Check all the other captions for missing first letter.

Line 407 – Figure 6 (left) compares NO? The authors want to say “Figure 7”

Line 445/453 – the authors want to say “Figure 6” instead of “Figure 7”.

Line 540 – what the authors want to say with “spells”? Use other word.

Author Response

We have replied to Reviewer 3's comments in the attached document
